# Interpretable AI Reasoning for the Identification of Vibrational Spectroscopic Markers of Acetaminophen Impurities

**Kaio Anderson dos Santos Soares**
Universidade Federal Rural do Semi-Árido (UFERSA)
Mossoró, RN, Brasil
kaio.soares@alunos.ufersa.edu.br

**Ludmilla da Silva Augusto**
Universidade Federal Rural do Semi-Árido (UFERSA)
Mossoró, RN, Brasil
ludmilla.augusto@alunos.ufersa.edu.br

**Roner Ferreira da Costa**
Universidade Federal Rural do Semi-Árido (UFERSA)
Mossoró, RN, Brasil
roner.costa@ufersa.edu.br

**Eveline Matias Bezerra**
Universidade Federal Rural do Semi-Árido (UFERSA)
Mossoró, RN, Brasil
eveline.bezerra@ufersa.edu.br

## Abstract

This study evaluates the capability of the Gemini 3 artificial intelligence model to identify and distinguish the active pharmaceutical ingredient Paracetamol (PCA) from its main synthetic impurities, p-aminophenol (PAP) and p-nitrophenol (PNP), based on reasoning grounded in vibrational spectroscopic data. To this end, vibrational mode tables obtained from theoretical calculations using Density Functional Theory (DFT) at the M06-2X/6-311 + +G(d,p) level of theory were employed. The model analyzed diagnostic markers distributed across three distinct spectral regions (200 to $4000 \text{ cm}^{-1}$), correlating specific structural variations with their corresponding vibrational signatures. The results demonstrate that Gemini 3 can associate topological and functional differences among molecules with characteristic spectroscopic patterns, yielding interpretable and consistent molecular representations. Consequently, this study highlights the potential of artificial intelligence models as auxiliary tools for automated pharmaceutical quality control, contributing to the reliable identification of impurities in synthetic drug pathways.

## Meaningfulness Statement

A meaningful representation of life in this work is one that preserves physically interpretable relationships among molecular structure, electronic properties, and spectroscopic response. By treating vibrational spectra obtained from first-principles calculations as meaningful representations, we demonstrate that an artificial intelligence model can reason over chemically grounded spectroscopic markers to distinguish an active pharmaceutical ingredient from structurally similar synthetic impurities. This approach goes beyond simple classification by aligning the model's internal representations with real scientific concepts, such as functional groups and electronic effects, thereby improving interpretability, reliability, and practical applicability in pharmaceutical quality control.

## 1 Introduction

Impurities in pharmaceutical products are undesired chemical species that may arise during synthesis, formulation, or storage processes, and their presence can directly compromise drug quality, safety, and therapeutic efficacy. According to their origin, these substances are commonly classified as organic impurities, inorganic impurities, or residual solvents (Liu (2024)). Regulatory agencies, therefore, require rigorous analytical strategies to ensure impurity levels remain within acceptable limits, reinforcing the importance of reliable identification and characterization methods in pharmaceutical quality control.

Paracetamol, also known as acetaminophen (PCA, N-(4-hydroxyphenyl)acetamide), is one of the most widely consumed analgesic and antipyretic drugs worldwide. Its pharmacological action primarily occurs in the central nervous system by inhibiting cyclooxygenase (COX) enzymes, which are responsible for prostaglandin synthesis and are directly associated with pain and fever mechanisms (Sherbiny & Wahba (2020)). Due to its extensive use, even trace amounts of impurities in PCA formulations represent a significant concern from both toxicological and regulatory perspectives.

PCA is commonly synthesized through an acetylation reaction between p-aminophenol (PAP) and acetic anhydride. Incomplete reactions or side pathways may result in the presence of synthetic impurities, such as residual PAP and p-nitrophenol (PNP). Among these, PAP is regarded as the primary impurity, since it is the starting material and is frequently detected in excess in its unreacted form (Sherbiny & Wahba (2020)). The presence of PNP, in turn, is particularly critical due to its distinct electronic effects and higher toxicological relevance. The molecular structures of PCA, PAP, and PNP are shown in Figure 1a, Figure 1b, and Figure 1c, respectively.

The occurrence of synthetic impurities poses a major challenge for pharmaceutical safety, as they can alter the balance between therapeutic benefits and potential risks. This scenario demands the implementation of robust analytical methodologies capable of distinguishing structurally similar compounds and ensuring drug purity (Chaugule et al. (2024); Tapkir et al. (2022)). Vibrational spectroscopic techniques, such as infrared and Raman spectroscopy, have proven to be powerful tools for this purpose, as they provide molecular fingerprints derived from the vibrational behavior of functional groups. When combined with theoretical simulations, these techniques allow direct comparison between experimental and calculated spectra, enhancing the reliability of molecular assignments and impurity identification (Tashiro (2022)).

In parallel, recent advances in artificial intelligence and machine learning have opened new perspectives for automated interpretation of spectroscopic data, enabling models to associate subtle spectral variations with underlying molecular structures. Within this context, the present study investigates the capability of the Gemini 3 artificial intelligence model to reason over vibrational spectroscopic representations derived from Density Functional Theory (DFT) calculations. By analyzing diagnostic vibrational markers of PCA and its main synthetic impurities (PAP and PNP), this work aims to evaluate the potential of AI-assisted spectroscopic reasoning as a complementary tool for pharmaceutical quality control, contributing to safer and more reliable identification of impurities in synthetic drug pathways.

## 2 METHODOLOGY

The workflow was designed to ensure the accurate determination of equilibrium geometries and the rigorous interpretation of their vibrational profiles. Initially, a systematic conformational search was performed for PCA and its impurities, PAP and PNP, using the Materials Studio software (Das (2020)) through the Conformers module. The purpose of this step was to explore the conformational space and identify the global minimum-energy structure for each compound, which served as the starting point for first-principles calculations.

In the subsequent step, the lowest-energy conformer of each chemical species was subjected to geometry optimization using Density Functional Theory (DFT) as implemented in the Gaussian09 software (Frisch et al. (2009)). The hybrid functional M06-2X was employed in conjunction with the $6\text{-}311++G(d,p)$ basis set to ensure a reliable description of electronic interactions and hydrogen bonding. All force and displacement convergence criteria were strictly satisfied. The absence of imaginary frequencies in the harmonic vibrational frequency calculations confirmed that the optimized geometries correspond to true minima on the potential energy surface.

The theoretical vibrational frequencies and their corresponding infrared intensities were extracted from the Gaussian output files using the Multiwfn software (Lu & Chen (2012)). Based on the extracted data, detailed vibrational mode assignment and analysis were carried out with the aid of the VEDA software (Jamróz (2004)) and Chemcraft (Che (2020)), which was used to visualize normal mode animations.

After constructing the vibrational assignment tables, the data were submitted for analysis using the Gemini 3 Pro model (reasoning mode). This final step aimed to evaluate the AI's ability to identify diagnostic spectroscopic markers that structurally distinguish the active pharmaceutical ingredient

Figure 1: Chemical structures of paracetamol-derived metabolites: (A) PCA – N-(4-hydroxyphenyl)acetamide ($C_8H_9NO_2$); (B) PAP – 4-aminophenol ($C_6H_7NO$); and (C) PNP – 4-nitrophenol ($C_6H_5NO_3$).

A) **PCA**

**IUPAC Name:** N-(4-hidroxifenil)acetamida
**Molecular Formula:** $C_8H_9NO_2$

B) **PAP**

**IUPAC Name:** 4-aminofenol
**Molecular Formula** $C_6H_7NO$

C) **PNP**

**IUPAC Name:** 4-nitrofenol
**Molecular Formula:** $C_6H_5NO_3$

PCA from its synthetic impurities PAP and PNP, thereby transforming vibrational assignment tables into meaningful representations for pharmaceutical quality control.

## 3   RESULTS AND DISCUSSION

All molecular geometries were successfully optimized, meeting all convergence criteria for energy, gradients, and atomic displacements. In addition, the absence of imaginary frequencies in the vibrational analysis confirms that the optimized structures correspond to true minima on the potential energy surface. The optimized geometries of PCA, PAP, and PNP are presented in Figure 2.

The evaluation of the Gemini 3 model's proficiency demonstrates that learning spectroscopic representations enables a clear binary distinction between the drug PCA and its precursor or degraded states (PAP and PNP), corroborating studies indicating that vibrational spectra act as true molecular fingerprints, sensitive to molecular topology and functional groups (Bal (2019)). The model was able to process the vibrational tables generated by DFT (M06-2X/6-311++G(d,p)) and identify how molecular topology dictates the individual spectroscopic signature of each compound, in agreement with works demonstrating the reliability of DFT methods in predicting and assigning vibrational modes of paracetamol and related compounds (Danten et al. (2006); Khadayat & Joshi (2024)). The Table 1 lists the spectroscopic markers identified in the analysis.

In the Fingerprint Region (200 to 1300 $cm^{-1}$), as shown in Figure 3, the model isolated low-frequency representations that are highly sensitive to the local electronic environment of the aromatic ring, a region widely recognized as crucial for the structural differentiation of organic compounds (Bal (2019)). The most significant feature for PNP was identified at 728 $cm^{-1}$ ($\pi$(O10C1O9N8)), corresponding to the out-of-plane deformation of the nitro group, a vibrational mode frequently associated with profound electronic alterations in the aromatic ring and with the compound's potential toxicity (Ren et al. (2021)). For PCA, the representation of complete acetylation is characterized by the mode at 984 $cm^{-1}$ ($\nu$(C10C9) + $\nu$(N8C9)), a structural signature absent in the PAP and PNP impurities, reinforcing previous results that associate specific bands in the fingerprint region with the presence of amide groups and the structural integrity of the drug (Danten et al. (2006); Khadayat & Joshi (2024)).

Figure 2: Representation of the molecular structures corresponding to the lowest-energy conformers of the compounds: (A) PCA, (B) PAP, and (C) PNP. Atom color scheme: oxygen (red), nitrogen (blue), and hydrogen (light gray). Carbon atoms are shown in pink for PCA, lilac for PAP, and green for PNP. Figures generated using the PyMOL software.

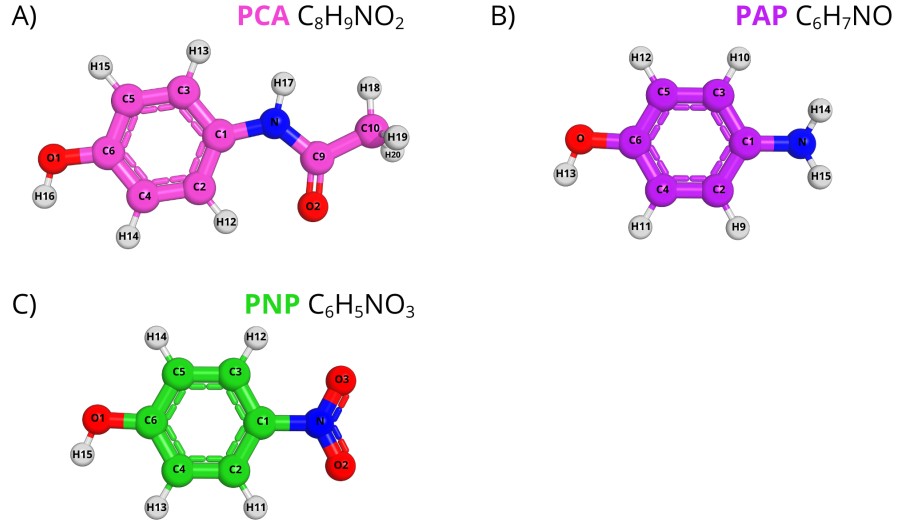

Table 1: Diagnostic vibrational representations and spectroscopic markers for purity differentiation between PCA, PAP, and PNP.

| Analytical Region | Freq. ($\mathrm{cm}^{-1}$) | Assignment (Normal Mode) | Mark. | Diagnostic Significance |
|---|---|---|---|---|
| Low Frequency | 728 | $\pi$(O10C1O9N8) | PNP | Out-of-plane def. (nitro group) |
| | 984 | $\nu$(C10C9) + $\nu$(N8C9) | PCA | Acetamide stretching |
| | 1093 | $\beta$(H14N8C1) | PAP | Primary amine def. |
| Mid-Infrared | 1462 | $\nu$(O10N8) + $\nu$(O9N8) | PNP | Symmetric stretching (nitro) |
| | 1576 | $\beta$(H17N8C9) | PCA | Amide II band (N−H deformation) |
| | 1660 | $\beta$(H15N8H14) | PAP | "Scissoring" def. (amine) |
| | 1807 | $\nu$(O11C9) | PCA | Amide I band (C=O carbonyl) |
| High Frequencies | 3671 | $\nu$(N8H17) | PCA | Secondary amide stretching |
| | 3697 | $\nu$(NH$_2$) | PAP | Primary amine stretching |
| | 3903 | $\nu$(O7H15) | PNP | Phenol under strong nitro effect |
| | 3910 | $\nu$(O7H16) | PCA | Standard phenolic hydroxyl |
| | 3924 | $\nu$(O7H13) | PAP | Phenol under amine effect |

The Functional Group Region (1300 to 1900 $\mathrm{cm}^{-1}$), as illustrated in Figure 4, proved to be the most discriminative feature space for artificial intelligence–assisted quality control, in agreement with studies indicating this range as the most informative for machine learning–based spectral classification (Kalatzis et al. (2025); Bikku et al. (2022)). The carbonyl stretching at 1807 $\mathrm{cm}^{-1}$ ($\nu$(O11C9)) in PCA represents the fundamental feature distinguishing the active drug from its PAP impurity, which is marked by the characteristic deformation of the primary amine at 1660 $\mathrm{cm}^{-1}$. The model correlated these frequencies with the transition from "impurity" to "safe drug" states, demonstrating that the presence of the Amide II band at 1576 $\mathrm{cm}^{-1}$ constitutes a necessary spectroscopic representation for pharmaceutical purity validation, as already observed in structural recognition protocols based on vibrational spectroscopy and machine learning (Ren et al. (2021)).

Finally, the analysis of the High-Frequency Region (2900 to 4000 $\mathrm{cm}^{-1}$), as shown in Figure 5, demonstrated how the model interprets subtle shifts as indicators of variations in molecular elec-

tronic density, a behavior widely documented in vibrational analyses of $O-H$ and $N-H$ bonds (Danten et al. (2006)). Phenolic stretching ($\nu(O!-!H)$) acts as a sensor of the chemical environment: whereas in PCA the signal at $3910$ $cm^{-1}$ represents an electronically stable configuration, the shift to $3903$ $cm^{-1}$ in PNP reveals the electron-withdrawing effect of the nitro group. The model's ability to discern variations of only a few wavenumbers reinforces its precision in identifying molecular representations that directly impact the safety and pharmacological efficacy, in line with recent studies demonstrating the high sensitivity of machine learning models to small spectral perturbations (Kalatzis et al. (2025); Bikku et al. (2022)).

Figure 3: Simulated infrared spectra of (a) PCA, (b) PAP, and (c) PNP obtained via DFT (M06-2X/6-311 + +G(d,p)) in the 200 to 1300 $cm^{-1}$ range. The intensities of PCA and PNP were scaled by a factor of $2\times$.

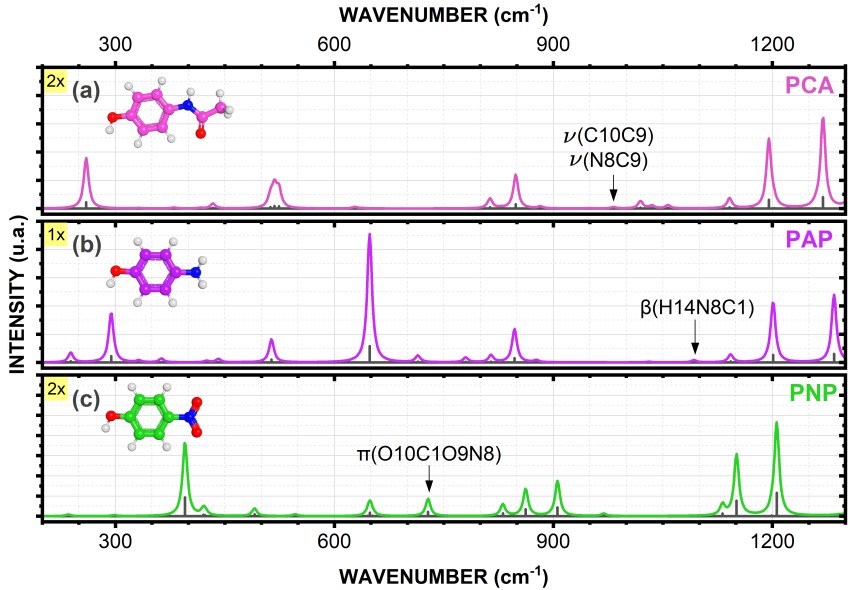

Figure 4: Simulated infrared spectra of (a) PCA, (b) PAP, and (c) PNP obtained via DFT (M06-2X/6-311 + +G(d,p)) in the 1300 to 1900 $cm^{-1}$ range.

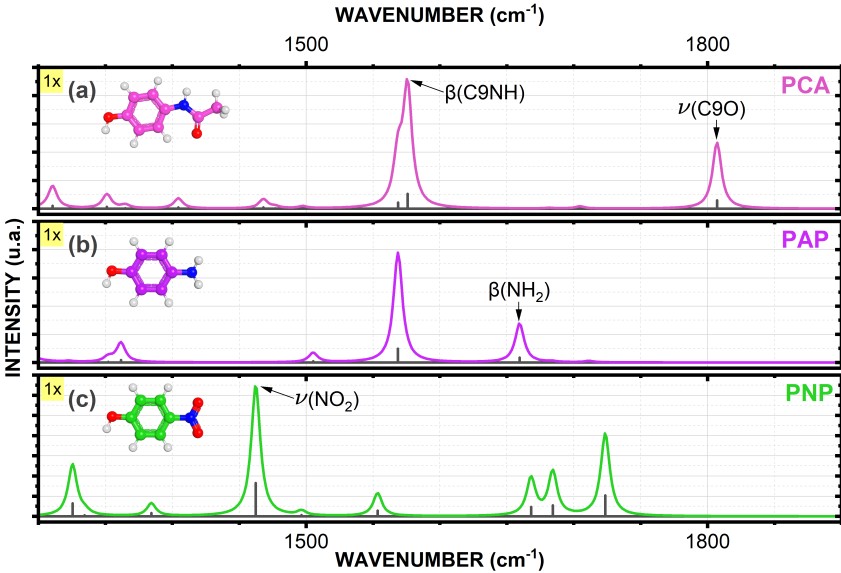

Figure 5: Simulated infrared spectra of (a) PCA, (b) PAP, and (c) PNP obtained via DFT (M06-2X/6-311 + +G(d,p)) in the $2900$ to $4000$ cm$^{-1}$ range. The intensities were scaled by a factor of $4\times$ for PCA and $3\times$ for PAP and PNP.

## 4  CONCLUSION

This study investigated the capability of the Gemini 3 artificial intelligence model to reason over vibrational spectroscopic representations derived from Density Functional Theory calculations in order to distinguish Paracetamol (PCA) from its main synthetic impurities, p-aminophenol (PAP) and p-nitrophenol (PNP). By combining first-principles vibrational analysis at the M06-2X/6-311 + +G(d,p) level with AI-based interpretative reasoning, the model demonstrated a consistent ability to associate molecular topology and functional group chemistry with diagnostic spectroscopic markers across the fingerprint, functional group, and high-frequency regions.

The results show that Gemini 3 successfully identified structurally meaningful vibrational features, such as the amide I and II bands characteristic of PCA, the nitro-group deformations exclusive to PNP, and the primary amine signatures associated with PAP. These findings reinforce the concept of vibrational spectra as robust molecular fingerprints and highlight the model's sensitivity to subtle frequency shifts induced by electronic and structural variations. Importantly, the AI did not merely classify the compounds but also provided interpretable reasoning grounded in chemically relevant spectral assignments, a critical requirement for applications in pharmaceutical quality control.

From an applied perspective, this work demonstrates the potential of artificial intelligence models as auxiliary tools for automated impurity identification, complementing rather than replacing traditional spectroscopic analysis. The integration of DFT-generated vibrational data with reasoning-capable AI systems opens new avenues for faster, more reliable, and more interpretable impurity screening in synthetic drug pathways, particularly in early-stage quality assessment and regulatory compliance workflows.

Future studies may extend this approach by incorporating experimental infrared and Raman spectra, expanding the impurity dataset, and benchmarking multiple AI architectures to assess generalization and robustness. Overall, the present results support the feasibility of AI-assisted spectroscopic reasoning as a promising strategy for enhancing pharmaceutical safety and analytical reliability.

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
