# OpenReview forum: "Interpretable AI Reasoning for the Identification of Vibrational Spectroscopic Markers of Acetaminophen Impurities"
_ICLR.cc/2026/Workshop/LMRL — ICLR 2026 Workshop LMRL Poster_

### Official Review · Reviewer_7Ca6 · 2026-02-16
**The paper,  Interpretable AI Reasoning for the Identification of Vibrational Spectroscopic Markers of Acetaminophen Impurities, combines standard DFT vibrational analysis with LLM-generated narrative interpretation. While the chemistry workflow seems technically sound, the machine learning contribution is very minimal. The study lacks benchmarking, generalization, quantitative evaluation, and proper impurity modeling.**

**Rating:** 4
**Confidence:** 3

**Review:**

Summary

This paper investigates the use of the Gemini 3 LMM to identify vibrational spectroscopic markers distinguishing paracetamol (PCA) from two synthetic impurities (p-aminophenol, PAP; p-nitrophenol, PNP), using Density Functional Theory (DFT), generated vibrational frequency tables as input. The authors perform conformational searches, DFT optimization, frequency calculations, and vibrational mode assignment, after which Gemini is prompted to reason over the resulting tables to highlight diagnostic spectral regions and functional-group-specific signatures. The work positions this workflow as a potential AI-assisted approach for pharmaceutical impurity identification and quality control.

While the study is clearly written and the underlying chemical evaluations are standard and sound, the machine learning component, from both methodological and applied aspects, seem to be very much limited.

Major concerns

1. The manuscript frames Gemini as an AI model that learns or reasons over spectroscopic representations. However, from my understanding, no training, fine-tuning, or actual representation learning is conducted in this work. Gemini seems to be used purely as a pretrained, general-purpose language model to interpret vibrational frequencies and assignments which are computed using standard quantum chemistry tools, and produces natural-language summaries linking known functional groups to known spectral regions. There is no evidence that Gemini learns new representations, extracts latent features, or performs beyond summarization and re-articulation of established spectroscopic knowledge.

It’s not fully clear why Gemini is required for this task at all. The manuscript does not demonstrate that Gemini adds capabilities beyond what could be achieved via rule-based analysis or conventional peak matching.

2. The paper doesn’t provide benchmarking against existing approaches for spectral classification or impurity detection. Comparisons to standard ML classifiers using accuracy or sensitivity metrics, robustness analyses, and cross-validation are required to evaluate Gemini’s performance.

3. The entire study focuses on one small, well-known molecule (acetaminophen) and two closely related impurities. This severely limits the generalizability of the conclusions. Moreover, acetaminophen is ubiquitous in chemistry literature and likely well represented in Gemini’s training data, raising concerns about implicit memorization rather than genuine reasoning.

4. Real impurity detection involves identifying low-concentration contaminants in mixtures. This work analyzes only pure compounds. There is no simulation of mixed spectra, mapping from spectral signatures to impurity concentration, evaluation of detection limits, connection between identified markers and quantitative toxicity thresholds.

Recommendation

To strengthen this work, the authors would need to:
- Evaluate multiple compounds across diverse chemical classes.
- Include quantitative benchmarks against conventional ML approaches.
- Demonstrate performance on mixture spectra and varying impurity levels.
- Clearly distinguish between LLM summarization and genuine learned representations.
- Justify the choice of a general-purpose LLM over domain-specific models.

---

### Official Review · Reviewer_Bcqd · 2026-02-25
**Scientifically Sound Application with Limited Representation Learning Evaluation**

**Rating:** 7
**Confidence:** 3

**Review:**

Summary

This paper investigates whether a reasoning-capable large language model (Gemini 3) can interpret vibrational spectroscopic data derived from Density Functional Theory (DFT) calculations to distinguish paracetamol (PCA) from two structurally related impurities (PAP and PNP). The study combines first-principles vibrational analysis (M06-2X/6-311++G(d,p)) with AI-based reasoning and argues that the model forms chemically meaningful representations aligned with functional groups and molecular topology.

Rather than performing standard classification, the work emphasizes interpretable reasoning over spectroscopic assignment tables across fingerprint, functional-group, and high-frequency regions.

Quality and Technical Soundness

The computational chemistry methodology is rigorous and clearly described. The conformational search, geometry optimization, verification of true minima (absence of imaginary frequencies), and vibrational mode assignment are appropriate and scientifically sound. The spectroscopic interpretation of functional groups (amide I/II bands, nitro-group deformation, primary amine signatures, etc.) is chemically consistent and well grounded in prior literature.

However, the machine learning evaluation lacks quantitative rigor. The manuscript does not provide:

Formal accuracy metrics

Baseline comparisons

Controlled evaluation protocols

Robustness or repeatability analysis

The reasoning capability of Gemini 3 is illustrated qualitatively but not systematically evaluated.

Originality and Significance

The work is original in framing vibrational spectra as meaningful structured representations and exploring whether a reasoning LLM can align with chemically grounded concepts. This is conceptually relevant to the workshop theme (Learning Meaningful Representations of Life).

The application to pharmaceutical impurity detection is practical and important. However, from a representation learning perspective, the novelty is limited since the paper evaluates an existing proprietary model rather than proposing or analyzing a new learning framework.

Clarity

The manuscript is clearly written and logically structured. The division of spectral regions (fingerprint, functional-group, high-frequency) is pedagogically effective. Figures and tables support the discussion well. The interpretability narrative is coherent and accessible.

Reproducibility would improve with inclusion of:

Exact prompting protocol

Model settings (temperature, number of runs)

Example full AI outputs

Pros

Strong and rigorous computational chemistry foundation

Clear interpretability framing aligned with scientific concepts

Practical pharmaceutical relevance

Well-structured spectral analysis

Cons

No quantitative evaluation of AI performance

No benchmarking against alternative models or baselines

Interpretability claims remain qualitative

Limited dataset (three molecules only)

Reproducibility details are insufficient

Suggestions for Improvement

Add quantitative evaluation (e.g., classification accuracy, confusion matrix).

Include multiple runs to assess consistency.

Compare with at least one baseline model (e.g., classical ML spectral classifier or smaller LLM).

Provide full prompting protocol for reproducibility.

Expand dataset to evaluate generalization beyond three compounds.

Overall Assessment

This is a scientifically solid and clearly presented application study demonstrating AI-assisted spectroscopic reasoning. The chemistry component is strong and the interpretability narrative is compelling. However, the machine learning validation lacks benchmarking and quantitative rigor.

For a workshop setting, the paper is relevant and potentially valuable, but additional systematic evaluation would significantly strengthen the contribution.

---

### Meta-Review · Area_Chair_h3M7 · 2026-02-28

**Recommendation:** Accept (Poster)
**Confidence:** 3

**Metareview:**

Accept.

---

### Decision · Program_Chairs · 2026-03-02

**Decision:**

Accept (Poster)

**Comment:**

Please see the meta-review.